# Unsupervised Homography Estimation on Multimodal Image Pair via Alternating Optimization

**Sanghyeob Song**[1,3]     **Jaihyun Lew**[1]     **Hyemi Jang**[2]     **Sungroh Yoon**[1,2*]

[1]Interdisciplinary Program in Artificial Intelligence, Seoul National University
[2]Department of Electrical and Computer Engineering, Seoul National University
[3]Samsung Electro-Mechanics
{songsang7, fudojhl, wkdal9512, sryoon}@snu.ac.kr

## Abstract

Estimating the homography between two images is crucial for mid- or high-level vision tasks, such as image stitching and fusion. However, using supervised learning methods is often challenging or costly due to the difficulty of collecting ground-truth data. In response, unsupervised learning approaches have emerged. Most early methods, though, assume that the given image pairs are from the same camera or have minor lighting differences. Consequently, while these methods perform effectively under such conditions, they generally fail when input image pairs come from different domains, referred to as multimodal image pairs.

To address these limitations, we propose AltO, an unsupervised learning framework for estimating homography in multimodal image pairs. Our method employs a two-phase alternating optimization framework, similar to Expectation-Maximization (EM), where one phase reduces the geometry gap and the other addresses the modality gap. To handle these gaps, we use Barlow Twins loss for the modality gap and propose an extended version, Geometry Barlow Twins, for the geometry gap. As a result, we demonstrate that our method, AltO, can be trained on multimodal datasets without any ground-truth data. It not only outperforms other unsupervised methods but is also compatible with various architectures of homography estimators. The source code can be found at: https://github.com/songsang7/AltO

## 1 Introduction

Homography is defined as the relationship between two planes when a 3D view is projected onto two different 2D planes. Many mid- or high-level vision tasks, such as image stitching [1], multispectral image fusion [2], and 3D reconstruction [3, 4], require low-level vision tasks such as image registration or alignment as preprocessing steps. Image registration is the process of aligning the coordinate systems of a given pair of images by estimating their geometric relationship. If the relationship between the image pair is a linear transformation, it is called homography.

Homography estimation for image alignment is known as reducing the geometric gap between a pair of images as depicted in Figure 1. It has been an active area of research since the pre-deep learning era, with prominent algorithms such as SIFT [5], SURF [6], and ORB [7]. The advent of deep learning ushered in the exploration of end-to-end approaches utilizing supervised learning, beginning with DHN [8]. These studies demonstrated the effectiveness of deep learning in homography estimation. However, supervised learning assumes the availability of ground-truth data for the relationship between image pairs or pre-aligned image pairs. The assumption is often unrealistic in real-world scenarios. Therefore, the rise of unsupervised learning has become essential to overcome these practical challenges.

---

*Corresponding Author

38th Conference on Neural Information Processing Systems (NeurIPS 2024).

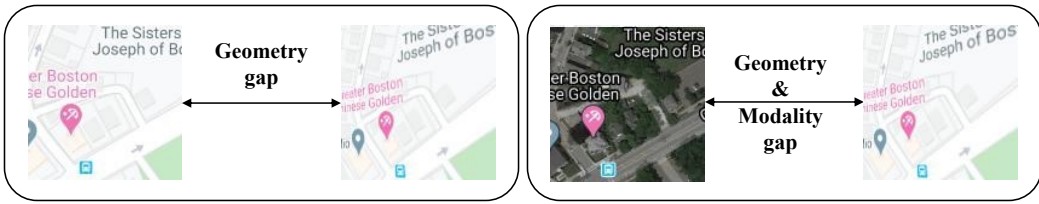

Figure 1: Examples of types of gaps. This paper will address both geometry and modality gaps simultaneously. Image pairs are introduced by DLKFM [11].

Driven by this necessity, various unsupervised learning methods for homography estimation have emerged. UDHN [9] and biHomE [10] estimate homography using only unaligned image pairs. These methods achieve comparable performance to existing supervised learning methods when applied to image pairs with the same modality or minor lighting differences. However, if there is a large modality gap, supervised learning methods still succeed in estimating homography due to the availability of ground-truth data, whereas unsupervised learning methods struggle to function effectively. Here, 'modality' refers to data domain, and 'modality gap' denotes differences in types of representation as shown in Figure 1. To address this modality issue, a simple approach that adding encoder for modality mapping can be considered. Yet, this approach leads to the trivial solution problem, which will be detailed in Section 3.2.

As a solution to all these problems, we propose a framework, named AltO, that handles each gap separately, without relying on ground-truth labels and avoiding the trivial solution problem. The proposed method leverages metric learning and utilizes two loss functions. The first is for the Geometry loss, which reduces the geometry gap by making images similar at the feature level. The second is the Modality loss, which guides the mapping of image pairs with different modalities into the same feature space. By repeatedly switching between the training of these two losses, our method achieves outstanding performance than other unsupervised learning-based methods on multimodal image pairs. Furthermore, our framework is independent of the registration network, so it can benefit directly from combining with a high-performance registration network.

In summary, the key contributions of this paper are:

- We propose AltO, an unsupervised learning framework for homography estimation on multimodal image pairs.

- We train the registration network by separately addressing the geometry and modality gaps through alternating optimization. In this process, we also introduce a new extended form of the loss function derived from the Barlow Twins loss [12].

- Our method has achieved superior performance both quantitatively and qualitatively compared to existing unsupervised learning-based approaches in multimodal conditions. Moreover, it has achieved performance nearly comparable to that of supervised learning.

## 2 Related Work

### 2.1 Hand-Crafted Homography Estimation

Homography estimation has advanced significantly with the development of the feature-based matching pipeline, especially after the introduction of SIFT [5]. This pipeline typically consists of feature detection, feature description, feature matching, and homography estimation. Various methods, such as SURF [6] and ORB [7], have been developed as variants for specific steps within this process. However, these methods do not account for the modality gap and are thus unsuitable for multi-modal environments. Recently, feature-based matching methods that handle multimodality, such as RIFT [13] and POS-GIFT [14], have been introduced. Despite these advances, their overall performance remains insufficient compared to learning-based methods. One of the main reasons is that most of them are based on HOG [15], which assumes planar cases without considering perspective effects.

## 2.2 Supervised Learning Homography Estimation

The first method to employ an end-to-end approach in the deep learning era is DHN [8]. This method trains a VGG [16]-based backbone network from a given pairs of images to predict offsets between four corresponding point pairs. The predicted offsets are then converted into a homography using the DLT [17] algorithm. Furthermore, the paper proposes a method for synthesizing datasets for training, which has been continually cited in subsequent research. Another supervised learning-based method is IHN [18], which, compared to DHN, uses a correlation volume to iteratively predict offsets, gradually reducing error. In addition, this method also experiments with a multimodal dataset and has demonstrated only minor error of alignment. This approach was also adopted in RHWF [19], which replaces some convolution layers with a transformer [20] blocks.

## 2.3 Unsupervised Learning Homography Estimation

Supervised learning requires ground-truth data, which is the homography between two images, for training. In practice, collecting such datasets, including the ground-truth labels, is nearly impossible. To overcome this limitation, unsupervised learning-based methods have emerged. The first proposed unsupervised learning method is UDHN [9]. It predicts offsets between a pair of input images, like DHN [8], and converts them into homography using the DLT [17]. It then warps one of the input images with this homography and compares it at the pixel level with the other image to measure similarity. Another method, biHomE [10], addresses the challenge of lighting variations through unsupervised learning. It utilizes a ResNet-34 [21] encoder that has been pre-trained on ImageNet [22], benefiting from the variety of lighting conditions in the dataset. This feature makes biHomE less susceptible to differences in lighting. However, it struggles when the input distribution significantly diverges from that of ImageNet, such as with multimodal image pairs. To address these cases, MU-Net [23] has been introduced and adopts CFOG [24]-based loss function. Although this demonstrates that CFOG can be integrated into an unsupervised learning framework and handle modality gap, CFOG still has weaknesses in handling geometric distortions such as rotation and scale.

# 3 Background

## 3.1 Homography

Homography is a 3×3 transformation matrix that maps one plane to another. It has a form of projection transformation and maps an arbitrary point $p$ in the image to $p'$ as follows:

$$p' \sim Hp \implies w \begin{bmatrix} x' \\ y' \\ 1 \end{bmatrix} = \begin{bmatrix} h_{11} & h_{12} & h_{13} \\ h_{21} & h_{22} & h_{23} \\ h_{31} & h_{32} & h_{33} \end{bmatrix} \begin{bmatrix} x \\ y \\ 1 \end{bmatrix} \tag{1}$$

In Equation (1), $H$ represents the homography matrix and the symbol $\sim$ denotes homogeneous coordinates, and $w$ indicates the scale factor that allows the use of equality in the equation. In the matrix H, $h_{13}$ and $h_{23}$ are related to translation, while $h_{11}$, $h_{12}$, $h_{21}$, and $h_{22}$ are associated with rotation, scaling, and shearing. Additionally, $h_{31}$ and $h_{32}$ are related to perspective effects and are set to 0 when perspective effects are not considered. Generally, when all effects are considered, it has 8 degrees of freedom (DOF), while in special cases where perspective effects are not considered, it has 6 DOF or fewer.

## 3.2 Trivial Solution Problem

As mentioned earlier, our goal is to address both the geometry and modality gaps together using unsupervised learning. One straightforward approach is to map each image into a shared space via an encoder and perform registration, similar to UDHN [9], using a reconstruction loss. Another approach, as in biHomE [10], places the encoder after the registration network, where registration is performed first, followed by loss calculation through the encoder. However, when all modules are trained simultaneously, both approaches collapse into a trivial solution: the encoder outputs a constant value, and the registration network predicts the identity matrix as the homography. It appears as if the encoder and registration network collaborate to minimize the loss in a trivial way. To avoid this, special techniques are needed to prevent collapse into trivial solutions.

### 3.3 Metric Learning

Metric learning refers to a method that trains models to increase the similarity between data points. Specifically, it is well-known within the field of self-supervised learning, which includes methods such as NPID [25], SimCLR [26], Simsiam [27], Barlow Twins [12], and VIC-Reg [28]. These methods commonly involve multiple input data and Siamese networks, and they focus on how to compare each output so that the model can learn good representations.

**Barlow Twins**, which we adopt, requires further explanation for the subsequent sections. Figure 2 illustrates the conceptual diagram of the Barlow Twins method. The encoder extracts the representations $r^A$ and $r^B$ from inputs $x^A$ and $x^B$, then produces embedding vectors $v^A$ and $v^B$ by passing these $r^A$ and $r^B$ through the projector. At this point, the dimension of both embedding vectors becomes $(N, D_v)$, where $N$ is the batch size and $D_v$ is the feature dimension. By calculating the similarity matrix between them, we obtain a matrix of size $(D_v, D_v)$. In

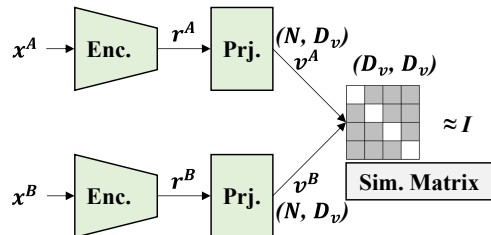

Figure 2: Conceptual Diagram of the Barlow Twins Method [12]

this process, the similarity is calculated using the correlation coefficient $C_{ij}$ in Equation (2), for the i-th element of $C$ from $v^A$ and the j-th element of $C$ from $v^B$. The bar over each vector ($\bar{v}^A$ and $\bar{v}^B$) denotes that the vectors have been mean-centered. The objective of the loss function $\mathcal{L}$ is to make the computed similarity matrix as close to the identity matrix as possible. This involves a balancing factor between the diagonal and off-diagonal elements. Especially, the term related with off-diagonal is called as 'Redundancy Reduction'. The formula is expressed as follows:

$$\mathcal{L} = \sum_i (1 - C_{ii})^2 + \lambda \sum_i \sum_{j \neq i} C_{ij}^2, \text{ where } C_{ij} = \frac{\sum_n (\bar{v}_{(n,i)}^A \bar{v}_{(n,j)}^B)}{\sqrt{\sum_n (\bar{v}_{(n,i)}^A)^2} \sqrt{\sum_n (\bar{v}_{(n,j)}^B)^2}} \tag{2}$$

## 4 Method

### 4.1 Alternating Optimization Framework

We present a visualized overview of our training framework at Fig. 3. Our objective is to train a registration network $\mathcal{R}$ that takes a moving image $I^A$ and a fixed image $I^B$ as input, each from modalities $A$ and $B$. It predicts a homography matrix $\hat{H}^{AB}$ that could warp ($\omega$) image $I^A$ to $\widetilde{I}^A = \omega(I^A, \hat{H}^{AB})$ which aligns with $I^B$.

Our training framework goes through two phases of optimization per mini-batch of data given, which are 'Geometry Learning' (GL) phase and the 'Modality-Agnostic Representation Learning' (MARL) phase. This framework optimizes alternatively, similarly to the Expectation-Maximization (EM) [29] algorithm and it can be expressed as follows:

$$\text{GL Phase}: \theta_t \leftarrow \underset{\theta}{\text{argmin}}[\text{GeometryGap}(\theta_{t-1}, \eta_{t-1}, \phi_{t-1})] \tag{3}$$

$$\text{MARL Phase}: \eta_t, \phi_t \leftarrow \underset{\eta, \phi}{\text{argmin}}[\text{ModalityGap}(\theta_t, \eta_{t-1}, \phi_{t-1})] \tag{4}$$

In expressions (3) and (4), $t$ represents the time step, and $\theta$, $\eta$, and $\phi$ denote the parameters of the registration network, encoder, and projector, respectively. The GL phase aims to maximize the similarity of local features of $\widetilde{I}^A$ and $I^B$, enabling the registration network to learn how to align the two images. The MARL phase is designed to learn a representation space that is modality-agnostic, so that the corresponding features of $\widetilde{I}^A$ and $I^B$ can optimally match. The two phases of optimization will be further detailed in the following sections.

### 4.2 Geometry Learning

In this phase of optimization, we fix $\eta$ and $\phi$, the weights of encoder $\mathcal{E}$ and projector $\mathcal{P}$, and then train the registration network $\mathcal{R}$. Our ultimate goal is to train the network $\mathcal{R}$ that predicts a homography

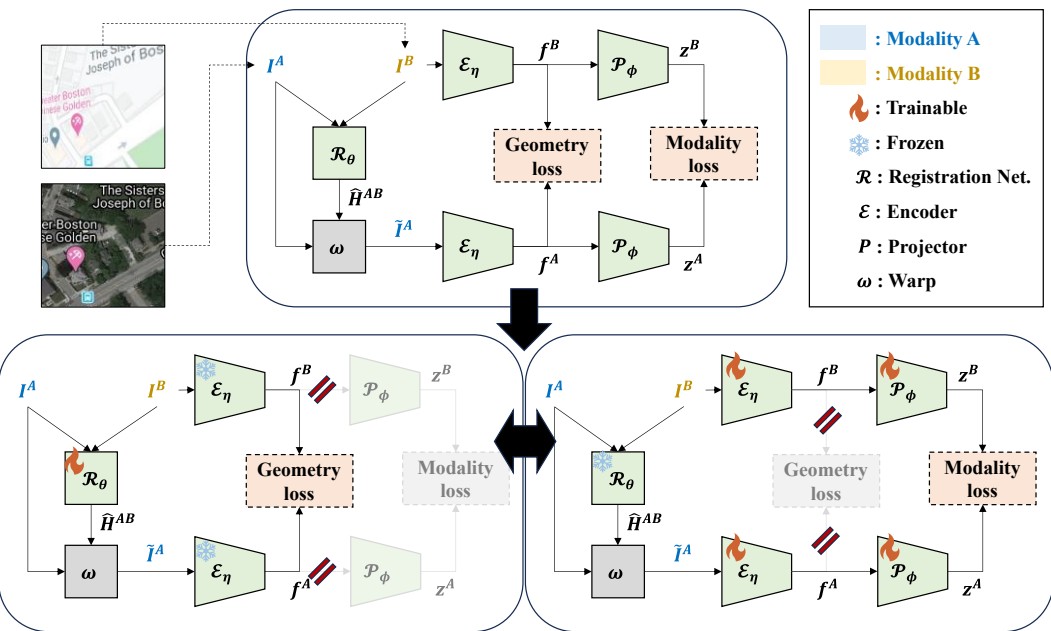

Figure 3: Overview of architecture. Upper diagram shows static view and lower diagrams illustrate phase switching between Geometry Learning (GL) phase and Modality-Agnostic Representation Learning (MARL) phase.

matrix $\hat{H}^{AB}$ which could warp a moving image $I^A$ to align with a fixed image $I^B$. Explicitly following this objective, we warp ($\omega$) image $I^A$ to acquire $\widetilde{I}^A = \omega(I^A, \hat{H}^{AB})$ and maximize the geometric similarity between the warped image $\widetilde{I}^A$ and the fixed image $I^B$. For the Geometry loss to use in training, we propose Geometry Barlow Twins (GBT) loss, a modified version of Barlow Twins [12] objective for 2-dimensional space. The features of two images $\widetilde{I}^A$ and $I^B$ are extracted with a shared encoder $\mathcal{E}$, and the output feature maps are used to compute the GBT loss. The GBT loss $\mathcal{L}_g$, is different from the original Barlow Twins in that we consider the spatial axis of the features as the batch dimension of the original Barlow Twins formula:

$$\mathcal{L}_g = \mathbb{E}_n\left[\sum_i(1 - C_{(n,ii)})^2 + \lambda\sum_i\sum_{j\neq i}C_{(n,ij)}^2\right],$$

$$\text{where } C_{(n,ij)} = \frac{\sum_{h,w}(\bar{f}_{(n,i,h,w)}^A\bar{f}_{(n,j,h,w)}^B)}{\sqrt{\sum_{h,w}(\bar{f}_{(n,i,h,w)}^A)^2}\sqrt{\sum_{h,w}(\bar{f}_{(n,j,h,w)}^B)^2}} \tag{5}$$

where $\bar{f}_{(n,i,h,w)}^A$ and $\bar{f}_{(n,i,h,w)}^B$ are feature vectors in $\mathbb{R}^{N\times D\times H\times W}$ corresponding to $\widetilde{I}^A$ and $I^B$, respectively, mean-normalized along the spatial dimensions, by subtracting the spatial mean from each unit. $n$ and $h, w$ each denote the batch and the spatial (horizontal and vertical) index and $i, j$ denote the index of channel dimension. Our objective can be considered as applying redundancy reduction on the spatial dimension of an image pair, maximizing the similarities of the local features at corresponding regions. By minimizing the geometric distance, or in other words, maximizing the geometrical similarity, the network $\mathcal{R}$ learns to geometrically align the given image pair. Visualization of our Geometry Learning (GL) phase can be found at the bottom-left of Figure 3.

## 4.3 Modality-Agnostic Representation Learning

Images of different modalities are susceptible to form a different distribution in the deep feature space. Maximizing the similarity of images from different modalities in such feature space may not lead to our desired result. Hence, for the model to successfully perform homography estimation given two images from different modalities, it should extract geometric information independent of modality. To help the GL phase to successfully work, the goal is to train an encoder which could

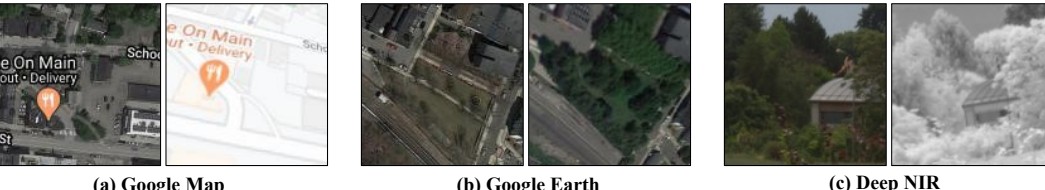

| (a) Google Map | (b) Google Earth | (c) Deep NIR |

Figure 4: Examples of image pair for each datasets. Google Map and Google Earth are introduced by DLKFM [11]. Deep NIR is proposed in [31]

form a modality-agnostic representational space of image features, so that our Geometry loss could properly work as intended. This process is referred to as Modality-Agnostic Representation Learning (MARL). During this phase, the registration network is fixed, and the encoder $\mathcal{E}$ and a projector $\mathcal{P}$ are trained. The standard form of the Barlow Twins loss [12] is adopted as the training loss, defined as follows:

$$\mathcal{L}_m = \sum_i (1 - C_{ii})^2 + \lambda \sum_i \sum_{j \neq i} C_{ij}^2, \text{ where } C_{ij} = \frac{\sum_n (\bar{z}_{(n,i)}^A \bar{z}_{(n,j)}^B)}{\sqrt{\sum_n (\bar{z}_{(n,i)}^A)^2} \sqrt{\sum_n (\bar{z}_{(n,j)}^B)^2}} \quad (6)$$

where $\bar{z}_{(n,i)}^A \in \mathbb{R}^{N \times D}$ and $\bar{z}_{(n,j)}^B \in \mathbb{R}^{N \times D}$ are the projected vectors of $f_{(n,i,h,w)}^A$ and $f_{(n,j,h,w)}^B$, respectively, mean-normalized along the batch dimension, by subtracting each unit with the batch-mean value. $n$ and $i, j$ denote the batch and the index of channel dimension. The Modality loss aims to constrain $z_{(n,i)}^A$ and $z_{(n,j)}^B$ to share a similar representational space, agnostic to the image's modality. To capture global feature information, we employ global average pooling (GAP) [21, 30], which removes the spatial dimensions to produce a feature vector. The necessity of GAP is discussed in Section 6.2.

Inspired by biHomE [10], the architecture uses the initial layers of ResNet-34 [21] as the encoder $\mathcal{E}$ and the latter layers as the projector $\mathcal{P}$. ResNet-34 comprises a stem layer, four stages, and a fully connected (FC) layer. Each stage halves the spatial resolution and contains 3 to 6 residual blocks. To retain finer structural details, we remove the max pooling layer after the first convolution in the stem layer. The stem layer and stages 1 and 2 serve as the encoder, while stage 3, along with GAP, functions as the projector. Section 6.4 confirms that this structure is the optimal division within ResNet-34. Stage 4 is optional, and the subsequent FC layer, excluding GAP, is omitted. As shown in Figure 3, encoders for $\tilde{I}^A$ and $I^B$ share weights, as do the projectors.

## 5 Experiments

### 5.1 Multimodal Datasets

**Google Map** is a multimodal dataset proposed in DLKFM [11]. It consists of pairs of satellite images and corresponding maps, which have different representation styles. There are approximately 9k training pairs and 1k test pairs of size $128 \times 128$.

**Google Earth** is another DLKFM dataset that provides multimodality by consisting of images of the same area taken in different seasons. The amount of data is about 9k for training and 1k for test. The input image size is also $128 \times 128$.

**Deep NIR** is a dataset proposed in [31]. It is an extension of the RGB-NIR scene dataset proposed by M. Brown et al. [32] through cropping and image-to-image translation. The RGB-NIR scene dataset consists of pairs of images, one captured by an RGB camera and the other by an NIR sensor at the same location. In this experiment, we use the oversampled $\times 100$ Deep NIR dataset. The dataset consists of approximately 14k training pairs and 2k test pairs. In this experiment, we resize these images to $192 \times 192$ and then apply dynamic deformation as proposed by DHN [8]. Finally, we obtain image pairs with a size of $128 \times 128$.

## 5.2 Implementation Details

**Registration networks** are selected to aim at demonstrating that our method can universally operate with various types of registration networks. To cover unique modules such as CNN, RNN, and Transformer [20], we chose DHN [8] (plain CNN), RAFT [33] (CNN+RNN), IHN [18] (CNN+iterative process), and RHWF [19] (Transformer+iterative process). In the case of DHN, following bi-HomE [10], the backbone network was switched from VGG [16] to ResNet-34 [21]. Additionally, IHN and RHWF employed a 1-scale network, which we refer to as IHN-1 and RHWF-1, respectively. In iterative process-based networks like RAFT, IHN, and RHWF, our AltO framework was applied at every time step, replacing the use of ground-truth labels. Further details are provided in the appendix.

**Experimental settings** of ours include using the PyTorch 1.13 library and an Nvidia RTX 8000 GPU with 48GB of VRAM for training each model. The models were optimized using the AdamW optimizer [34], a one-cycle learning rate schedule [35], a maximum learning rate of 3e-4, and a weight decay of 1e-5. Additionally, gradient clipping of ±1.0 is applied during the backward pass for the Geometry loss. The training protocol repeats for a total of 200 epochs with a batch size set to 16. Regarding the loss parameters, $\lambda$ is set to 0.005, consistent with standard settings of Barlow Twins [12].

## 5.3 Evaluation Metric

The evaluation metric used is Mean Average Corner Error (MACE). Corner error is the Euclidean distance between the positions warped by the correct homography $H^{AB}$ and the predicted homography $\hat{H}^{AB}$ for a corner of $I^A$. ACE is the average corner error across four corners, and MACE is the overall mean of ACE across the dataset.

$$\text{MACE}(H^{AB}, \hat{H}^{AB}) = \mathop{\mathbb{E}}_{n \in N} \left[ \mathop{\mathbb{E}}_{c \in \mathcal{C}} \left[ ||\omega(c, H^{AB}) - \omega(c, \hat{H}^{AB})||_2 \right] \right] \tag{7}$$

In Equation (7), $\mathcal{C}$ is the set of corner points of $I^A$ and $N$ is the set of samples. In this evaluation, a lower MACE indicates less error and better performance.

Table 1: Experimental results on three benchmarks, Google Map [11], Google Earth [11] and Deep NIR [31]. We report the Mean Average Corner Error (MACE) for evaluation. Our approach shows state-of-the-art performance among unsupervised methods, with robust compatibility across diverse network architectures for homography estimation.

| Learning Type | Method | Google Map [11] | Google Earth [11] | Deep NIR [31] |
|---|---|---|---|---|
| | (No warping) | 23.98 | 23.76 | 24.75 |
| Supervised | DHN [8] | 4.00 | 7.08 | 6.91 |
| | RAFT [33] | 2.24 | 1.9 | 3.34 |
| | IHN-1 [18] | 0.92 | 1.60 | 2.11 |
| | RHWF-1 [19] | 0.73 | 1.40 | 2.06 |
| Unsupervised | UDHN [9] | 28.58 | 18.71 | 24.97 |
| | CAU [36] | 24.00 | 23.77 | 24.9 |
| | biHomE [10] | 24.08 | 23.55 | 26.37 |
| | DHN [8] + AltO (ours) | 6.19 | 6.52 | 12.35 |
| | RAFT [33] + AltO (ours) | 3.10 | 3.24 | 3.60 |
| | IHN-1 [18] + AltO (ours) | **3.06** | **1.82** | **3.11** |
| | RHWF-1 [19] + AltO (ours) | 3.49 | 1.84 | 3.22 |

## 5.4 Evaluation Results

The experimental results are shown in Table 1. When image pairs are unaligned, or if the identity matrix is used as the homography, MACE typically exceeds 23 px. Thus, any method yielding MACE above this threshold can be considered unsuccessful in training. Most conventional unsupervised methods fail to train effectively on the benchmark datasets. UDHN [9], which directly compares $I^B$ and $\tilde{I}^A$ in pixel space, is highly sensitive to modality changes, resulting in failed training on most datasets except Google Earth [11]. Although the Google Earth dataset is multimodal, its image pairs have similar intensity distributions, allowing only limited training in pixel space. Similarly, CAU [36], which addresses only minor lighting variations, suffers from modality gaps and large displacements,

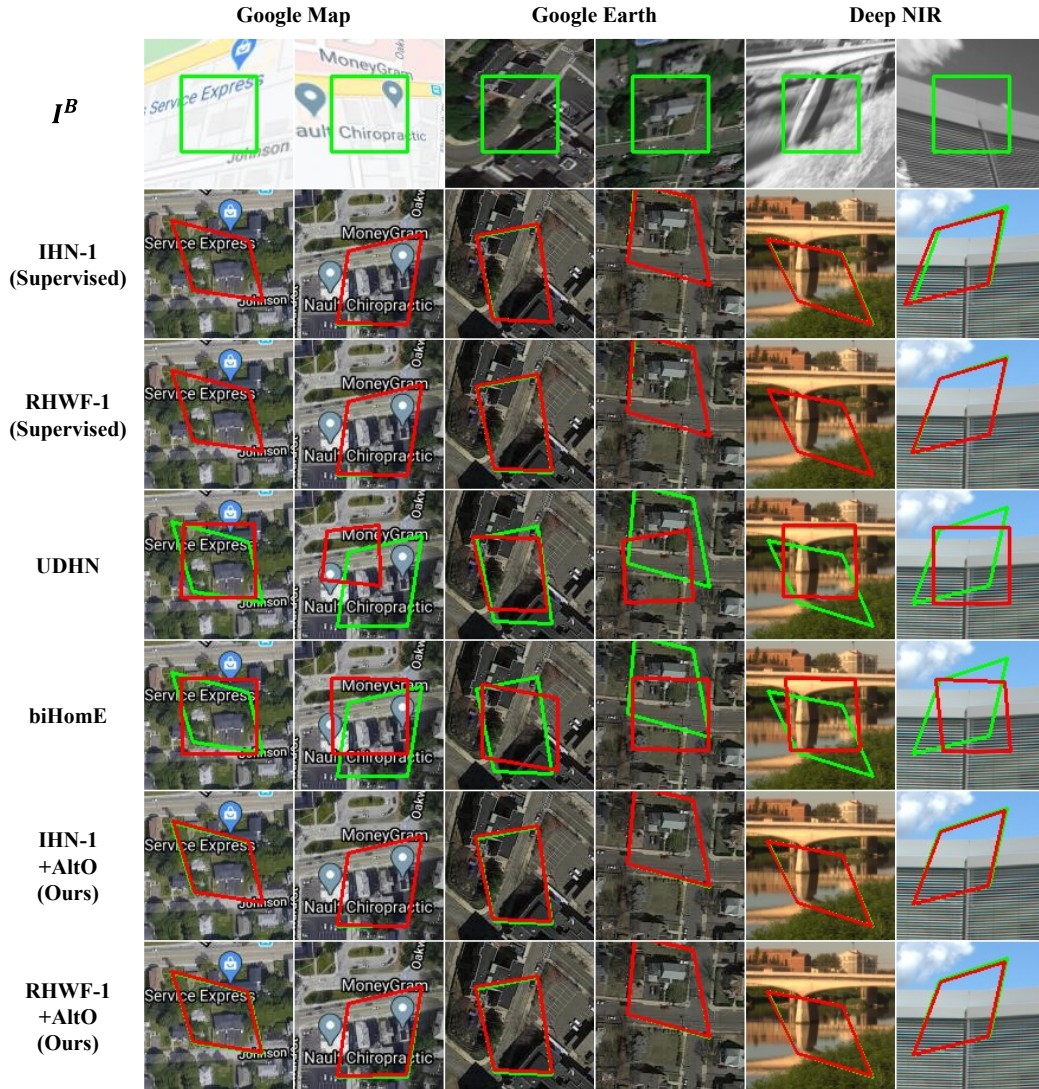

Figure 5: Visualization of homography estimation using center box. The first row shows the state before applying homography (green rectangles). Subsequent rows compare the results after applying ground-truth (green) and predicted (red) homography matrices. Our method, AltO, closely matches supervised learning-based methods, while other unsupervised approaches underperform.

leading to consistently low performance across all datasets, including Google Earth. biHomE [10], structurally similar to our method, uses an ImageNet [22] pre-trained ResNet-34 [21] as the encoder, which struggles with images outside the ImageNet distribution.

In contrast, our proposed method achieves significantly higher performance than existing unsupervised approaches. Additionally, it is compatible with any registration network architecture, $\mathcal{R}$, showing only a marginal performance gap compared to supervised training. Although this gap may appear quantitatively large, it would be negligible in practical applications. Figure 5 illustrates the qualitative results, showing boxes warped using ground-truth and estimated homography matrices. While conventional unsupervised methods fail to align boxes, our method achieves high alignment accuracy, with only minor differences compared to supervised methods using the same $\mathcal{R}$. These results indicate that our method, AltO, can replace supervision from ground-truth homography matrices.

In the appendix, there are additional qualitative results with another way of demonstration. There are comparisons of $I^A$, $I^B$, and $\widetilde{I}^A$ for each method.

# 6 Ablation Study

## 6.1 The Effectiveness of Alternating

In our AltO, the key technique is alternating, designed to address the trivial solution problem from unintended collaboration, as discussed in Section 3.2. This ablation study demonstrates its effectiveness by comparing

Table 2: MACE for methods with and without alternating and MARL

| Method | No Alternating | | Alternating |
| | with MARL | w/o MARL | with MARL |
|---|---|---|---|
| DHN [8] + AltO | 24.09 | 24.27 | **6.19** |
| RAFT [33] + AltO | 26.21 | 25.91 | **3.10** |
| IHN-1 [18] + AltO | 24.37 | 23.14 | **3.06** |
| RHWF-1 [19] + AltO | 18.88 | 24.07 | **3.49** |

cases with and without alternating. The experiment was conducted on the Google Map dataset [11], measuring the MACE values described in Section 5.3. Table 2 presents the experiment conducted with and without Modality-Agnostic Representation Learning (MARL) when alternating is absent. As expected, the result shows that effective learning occurs only with alternating.

## 6.2 The Necessity of Global Average Pooling

In Section 4.3, we stated that global average pooling (GAP) [30] is applied at the end of the projector $\mathcal{P}$. This ablation study compares performance with and without GAP. The experiment was conducted using the Google Map dataset [11], with the result presented in Table 3. As shown by the result, effective learning occurs only with GAP. This is because, without GAP, local features are forced to be

Table 3: MACE for methods with and without global average pooling (GAP) [30].

| Method | w/o GAP | with GAP |
|---|---|---|
| DHN [8] + AltO | 24.07 | **6.19** |
| RAFT [33] + AltO | 24.07 | **3.10** |
| IHN-1 [18] + AltO | 24.01 | **3.06** |
| RHWF-1 [19] + AltO | 24.08 | **3.49** |

similar by comparing fine details before the registration network is fully trained, which leads to the trivial solution problem discussed in Section 3.2.

## 6.3 Combination of Loss Functions

Our proposed AltO requires both Geometry and Modality losses. While we suggest using Barlow Twins [12] for both, this section examines the performance impact of alternative losses. For the experiments, the registration network $\mathcal{R}$ is set to IHN-1 [18], and the dataset used is Google Map [11]. We compare with Info NCE [37] and VIC-Reg [28], widely used contrastive losses in Siamese self-supervised learning, and additionally test Mean Squared Error (MSE) for the Geometry loss. Modality loss is used in its original form, while Geometry loss is expanded to include spatial dimensions, as described in Section 4.2. In this case, Info NCE is introduced as Patch NCE in CUT [38].

Table 4: Ablation study on Geometry and Modality losses, exploring all combinations of three popular contrastive losses and MSE for Geometry loss.

| Geometry loss | Modality loss | MACE |
|---|---|---|
| Barlow Twins [12] | Barlow Twins | **3.06** |
| | Info NCE [37] | 3.73 |
| | VIC-Reg [28] | 3.36 |
| Patch NCE [38] | Barlow Twins | 3.56 |
| | Info NCE | 4.71 |
| | VIC-Reg | 3.4 |
| VIC-Reg | Barlow Twins | 21.47 |
| | Info NCE | 4.95 |
| | VIC-Reg | 23.99 |
| MSE | Barlow Twins | 3.74 |
| | Info NCE | 3.97 |
| | VIC-Reg | 3.18 |

Table 4 presents the MACE results for different combinations of loss functions. It is known from SimCLR [26] that using Info NCE or Patch NCE benefits from a larger number of contrasting instance pairs. However, in our implementation, the batch size affecting the Modality loss is limited to 16, and the spatial resolution impacting the Geometry loss is only 32 by 32 (1024 pairs), which is relatively small. This likely leads to a decline in performance. Additionally, VIC-Reg exhibited instability due to the three separate balance factors required for its components: variance, invariance, and covariance. MSE is another choice that could be used as a distance metric to compute the geometrical gap. This option of loss leads to a fair performance, but short in comparison to our

Barlow Twins-based objective. We hypothesize the reason for this observation as below: even if image pairs are at corresponding locations, the information at those locations may not have identical feature representations. For example, while one image may include detailed edge descriptions, the other might lack such details due to differences in representation style. This difference in information quantity results in discrepancies in the embedded features. Essentially, while features at corresponding locations should be similar, they cannot be exactly the same for this reason.

In contrast, our proposed Barlow Twins-based losses do not suffer from the weaknesses present in other losses. First, unlike Info NCE and Patch NCE, they do not require a large number of contrasting pairs, as shown in the original Barlow Twins paper [12]. Additionally, they use only one balancing factor, making them significantly more stable than VIC-Reg. Lastly, unlike MSE, they do not require the embedded features to be exactly identical but instead aim to increase similarity, which is more applicable to real-world scenarios. These advantages allow our proposed combination to achieve superior performance.

### 6.4 Architecture of Encoder and Projector

As mentioned in Section 5.2, we have adapted ResNet-34 [21] by dividing it into an encoder and a projector. In doing so, there were several cases regarding which stages to use as the encoder and which to designate as the projector. This ablation study aims to measure the performance for each of these configurations. The Google Map dataset [11] is used, with IHN-1 [18] as the registration network $\mathcal{R}$. The indices used in Table 5 represent the sequential order of ResNet stages.

Table 5: Encoder and projector configurations within ResNet-34 [21]. Each number represents a ResNet stage; all encoders include the stem layer, and all projectors end with GAP [30].

| Encoder (Stage No.) | Projector (Stage No.) | MACE |
|---|---|---|
| 1 | 2, 3 | 3.82 |
| 1 | 2, 3, 4 | 3.78 |
| 1,2 | 3 | **3.06** |
| 1,2 | 3, 4 | 3.07 |
| 1,2,3 | 4 | 9.76 |

The result indicates that the encoder performs best when at least two stages are utilized. While using three stages can enhance the encoder's mapping capabilities, the resolution is halved, leading to decreased precision. Conversely, using only one stage results in better resolution compared to two stages, but the mapping capabilities are inadequate. The experiments reveal that the sweet spot between resolution and mapping ability trade-offs is achieved with two stages. For the projector, using stages beyond the third does not significantly affect performance, except when only the shallowest, the fourth stage, is used.

## 7 Conclusion

We propose AltO, a new learning framework, capable of training on multimodal image pairs through unsupervised learning. By alternating optimization, like Expectation-Maximization [29], AltO handles both geometry and modality gaps separately. This framework employs two types of loss functions, using the Barlow Twins [12] and its extended version, to effectively address both gaps. Experimental results demonstrate that the registration network is stably trained within our framework across various backbones. Furthermore, it outperforms other unsupervised learning-based methods in MACE evaluations and achieves performance close to that of supervised learning-based methods across multiple datasets.

## 8 Limitation and Future Work

Despite its advantages, our method encounters some limitations. Firstly, applying the same registration network results in slightly reduced performance compared to supervised learning, which directly utilizes ground-truth labels. Secondly, our framework requires training additional modules to replace ground-truth labels, which slows down the training process.

To address these limitations, future work could explore new designs, such as integrating Transformers [20] into the encoder and projector to improve performance. Additionally, reducing training time is essential, motivating efforts to develop a single-phase framework that avoids the trivial solution problem mentioned in Section 3.2.

## 9 Acknowledgements

- This work was supported by Samsung Electro-Mechanics.

- This work was supported by Institute of Information & communications Technology Planning & Evaluation (IITP) grant funded by the Korea government(MSIT) [NO.RS-2021-II211343, Artificial Intelligence Graduate School Program (Seoul National University)]

- This work was supported by the BK21 FOUR program of the Education and Research Program for Future ICT Pioneers, Seoul National University in 2024.

- This work was supported by the National Research Foundation of Korea (NRF) grant funded by the Korea government (MSIT) (No. 2022R1A3B1077720).

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

# A Appendix

## A.1 Additional Visualization of the Results from the Main Experiment

We provide additional results of our method and baseline methods for comparison as below.

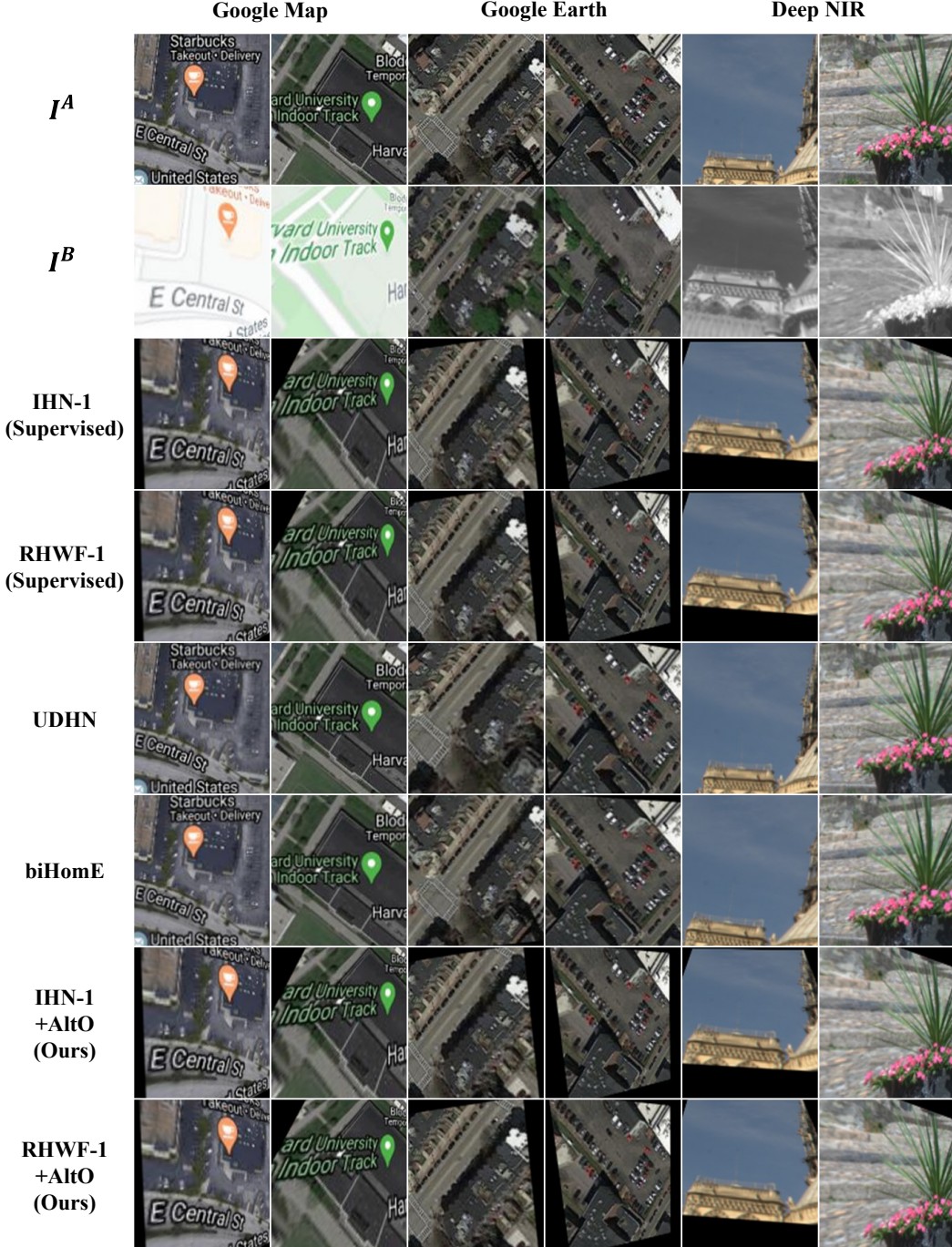

Figure 6: Moving, fixed, and warped images. The first row displays the moving image ($I^A$), the second row shows the fixed image ($I^B$), and the remaining rows present the warped images ($\tilde{I}^A$) derived from $I^A$ for each method.

## A.2 Integration of Iterative Registration Networks into AltO

In this section, we provide a detailed explanation of integrating a registration network with an iterative process into AltO. In the main experiment, we used DHN [8], RAFT [33], IHN-1 [18], and RHWF-1 [19] as the registration network $\mathcal{R}$. Among these, RAFT, IHN-1, and RHWF-1 employ an iterative process, predicting intermediate homographies over multiple time steps for each homography estimation. When supervised learning is applied, the loss function is used for homographies output at each time step. Since AltO is also intended as a replacement for supervision, we similarly apply the Geometry loss to the homography output at each time step. For RAFT, originally designed for optical flow estimation, we implicitly incorporate DLT [17] to convert optical flow into homography. Figure 7 illustrates this process.

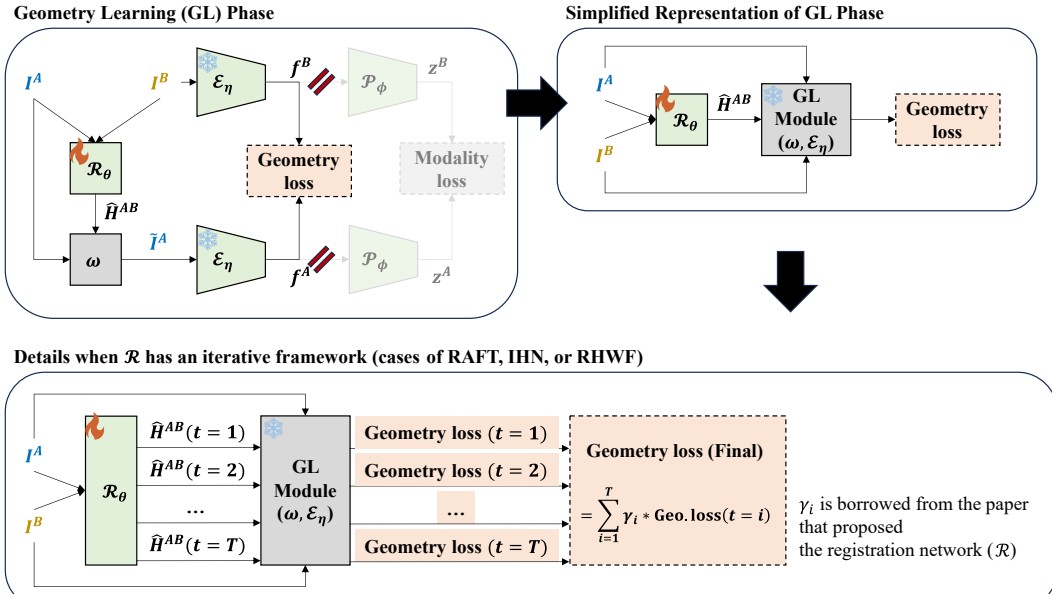

Figure 7: Visualization of the iterative registration process in AltO. For RAFT [33], DLT [17] is implicitly used to transform optical flow into homography across time steps.

