# OpenReview forum: "Unsupervised Homography Estimation on Multimodal Image Pair via Alternating Optimization"
_NeurIPS.cc/2024/Conference — NeurIPS 2024 poster_

### Official Review · Reviewer_EsVS · 2024-06-26

**Soundness:** 2
**Presentation:** 2
**Contribution:** 2
**Rating:** 3
**Confidence:** 5

**Summary:**

The paper proposes an unsupervised homography estimation method for multimodal image pairs using an alternating optimization approach. The claimed key innovation is the introduction of the Geometry Barlow Twins loss function for the alternating optimization. The authors show that their approach works on 3 multimodal datasets and different homography estimation architecutres.

**Strengths:**

The alternating optimization framework together with Geometry Barlow Twins loss seem to be a fresh perspective on unsupervised multimodal homography estimation.

**Weaknesses:**

Weaknesses
1. Discussion on the Feasibility and Rationality of the Proposed Method: First, for unsupervised training of networks based on iterative prediction, such as RAFT, to ensure stability during training, related methods [1-2] typically apply some form of direct supervision to the motion predicted by the network. This is different from the approach proposed in this paper, which only uses the Geometry Barlow Twins loss for brightness supervision. Second, how RAFT can be used for homography estimation should also be explained, because it is designed for optical flow estimation. Moreover, the paper does not explain how the proposed Geometry Barlow Twins loss supervises the intermediate stages of iterative prediction, whereas RAFT, IHN, and RHWF, along with methods leveraging their structures [1-2], generally provide details on their supervision mechanisms on the intermediate stages. This raises concerns about the feasibility of the proposed supervision method in this paper. Additionally, the effectiveness of the Modality-Agnostic Representation Learning (MARL) introduced in section 4.3 is questionable because it lacks spatial information in its supervision. As mentioned in section 3.2, the projector removes spatial information from the feature maps. The authors should provide a convincing and thorough explanation for these issues.

2. Doubt about the Effectiveness of the Proposed Method: For example, the paper proposes the alternating optimization (AltO) method but does not provide sufficient experimental results to demonstrate its superiority over other strategies, such as directly cascading all the modules. Furthermore, the paper lacks a comparative demonstration of the features extracted with and without the MARL phase, making the advantages of introducing this phase less convincing.

3. Insufficient Experimental Validation: The paper conducts experiments on only 3 cross-modal datasets, among which only the GoogleMap dataset exhibits significant modality differences. The GoogleEarth dataset mainly consists of images taken in different seasons [3]. Part of the DeepIR dataset is simulated multispectral data [4], which will significantly reduce the difficulty of homography estimation. It would be beneficial to conduct experiments on more challenging multimodal datasets, such as those involving VIS-SAR modalities.

[1] Stone, A., Maurer, D., Ayvaci, A., Angelova, A., & Jonschkowski, R. (2021). Smurf: Self-teaching multi-frame unsupervised raft with full-image warping. In Proceedings of the IEEE/CVF conference on Computer Vision and Pattern Recognition (pp. 3887-3896).
[2] Liang, Y., Liu, J., Zhang, D., & Fu, Y. (2023). Mpi-flow: Learning realistic optical flow with multiplane images. In Proceedings of the IEEE/CVF International Conference on Computer Vision (pp. 13857-13868).
[3] Zhao, Y., Huang, X., & Zhang, Z. (2021). Deep lucas-kanade homography for multimodal image alignment. In Proceedings of the IEEE/CVF conference on computer vision and pattern recognition (pp. 15950-15959).
[4] Sa, I., Lim, J. Y., Ahn, H. S., & MacDonald, B. (2022). deepNIR: Datasets for generating synthetic NIR images and improved fruit detection system using deep learning techniques. Sensors, 22(13), 4721.

**Questions:**

Please refer to the Weaknesses.

**Limitations:**

The paper discusses the additional training cost arising from the inclusion of an additional module in the two-phase network, and explores potential solutions for addressing this issue in future research. However, the method’s generalization capabilities are not thoroughly explored, with experimental datasets limited to satellite images, maps, RGB, and NIR images. Future research could involve testing the method on a broader range of datasets to validate its generalization capabilities.

---

> ### Author Rebuttal · Authors · 2024-08-06
>
> Thank you for taking the time to review our work. We apologize for the lack of detailed explanations regarding the proposed method. Below are some additional clarifications:
>
> **Weakness 1.**
>
> **Weakness 1.1.** *No other direct supervisions*
>
> Increasing the similarity of local features between two inputs is sufficient and equivalent to estimating homography. This is because homography estimation assumes a linear transformation between pairs of static scenes. Therefore, smoothness is automatically satisfied, and global motion is equivalent to local motion and homography.
>
> **Weakness 1.2.** *How is AltO applied to iterative frameworks such as RAFT, IHN, or RHWF?*
>
> We used each prediction from the registration network to individually calculate the GBT loss. Then, all GBT losses at each time step were combined by weighted summation, similar to the supervised learning situation. The weights used are also borrowed from the supervised learning case. The only difference from the supervised case is that AltO is used instead of the ground truth. If we had not effectively utilized the full capability of the registration networks, we would not have been able to achieve such performance.
>
> In Figure R1 of the attached rebuttal PDF, there is an illustration of the explained details, so please refer to it.
>
> **Weakness 1.3.** *How can RAFT be used for homography estimation?*
>
> A motion flow map from an optical flow network, such as RAFT, can be regarded as HW-correspondences (H: height, W: width). If HW is 4 or larger, the Direct Linear Transform (DLT) algorithm can fit a single homography. This homography best approximates the flow map in terms of least squared error.
>
> Generally, in homography estimation, which has 8 degrees of freedom, most algorithms predict only 4 correspondences at the corners of images. They then convert these correspondences to a homography using DLT. Therefore, there is no reason HW-correspondences cannot be used for estimating homography.
>
> If you are concerned about secondary aspects of optical flow, such as smoothness or the correction between global camera motion and local motion, please refer to our response in Weakness 1.1. As we explained there, homography estimation is estimating a linear transformation between pairs of static images. Therefore, these issues do not arise.
>
> **Weakness 1.4.** *What is the effectiveness of Modality-Agnostic Representation Learning (MARL)?*
>
> It is to train the encoder that maps input images to the same feature space, not to train registration network directly. This goal allows our geometry loss to function properly regardless of modality difference. To achieve this goal, the loss term of MARL should be designed to enhance global similarity, rather than the local similarity of corresponding points, as is done with the GBT loss. This is why the global average pooling (GAP) layer should be included at the end of the projector.
>
> Table R2 shows the training results on Google Map dataset when GAP is not applied and spatial information is preserved to compute the similarity of local correspondences. It can be observed that proper training is not achieved in this case because local similarity is already met by the GBT loss, without considering global similarity.
>
> Table R2: Ablation results for including and excluding GAP.
> | Method | without GAP | with GAP |
> |--------|:-----------:|:--------:|
> | DHN + AltO | 24.07 | **6.19** |
> | RAFT + AltO | 24.07 | **3.10** |
> | IHN + AltO | 24.01 | **3.06** |
> | RHWF + AltO | 24.08 | **3.49** |
>
> **Weakness 2.** *Effectiveness of the Proposed Method. (Why is alternating necessary?)*
>
> To prevent unintended collaborations and a collapse into trivial solutions, we introduced an alternating training strategy. Training the encoder and registration network together end-to-end without alternating can cause the encoder to output a constant value and the registration network to always output the identity matrix as the homography. Alternating training isolates these modules. This ensures the encoder maps input images to the same feature space, and the registration network estimates the true homography between the two images.
>
> Table R1 in the Auther Rebuttal (above) shows the experimental results when alternating training is not applied, demonstrating that proper training does not occur.
>
> **Weakness 3. & Limitation.** *Insufficient Experimental Validation*
>
> You pointed out that the difficulty of the DeepNIR dataset is reduced due to the presence of simulated pairs. So we conducted experiments on the 'RGB-NIR' [1] dataset, which consists purely of RGB and NIR pairs without any synthetic images. We selected 'IHN' as the registration network, as it represents our primary model. The experimental results, as shown in Table R3 below, indicate that all methods performed better on this dataset compared to DeepNIR. This suggests that the DeepNIR dataset is harder. We believe this is because the DeepNIR dataset requires the model to learn two distributions (real NIR and synthetic NIR images), whereas the RGB-NIR dataset only requires learning a single distribution (real NIR).
>
> Additionally, we conducted experiments on the 'OS Dataset' [2], which is a VIS-SAR type dataset, as per your suggestion. The results are also presented in Table R3.
>
> In conclusion, our method outperformed other unsupervised methods on both of the additional datasets.
>
> Table R3: MACE evaluation on two datasets.
> | Method | RGB-NIR | OS Dataset |
> |:-------|:-------:|:----------:|
> | IHN (+Supervision) | 1.57 | 6.33 |
> | UDHN | 24.30 | 32.93 |
> | CA-UDHN | 24.27 | 24.96 |
> | biHomE | 25.18 | 24.99 |
> | IHN + AltO | **2.66** | **14.12** |
>
> [1] Matthew Brown et al., Multi-spectral sift for scene category recognition. CVPR 2011.
>
> [2] Xiang, Yuming, et al., Automatic registration of optical and sar images via improved phase congruency model, IEEE Journal of Earth Observations and Remote Sensing, 2020

---

> ### Author Response · Authors · 2024-08-12
> **Looking forward to your post-rebuttal comment!**
>
> Dear Reviewer EsVS
>
> Thank you once again for participating in the review process and for providing such thoughtful feedback. We wanted to kindly follow up regarding the rebuttal we submitted. We understand that this is a busy time, and we greatly appreciate your efforts. To summarize our rebuttal to your review:
>
> * Provided detailed answers to all the questions you raised.
> * Conducted a new experiment to demonstrate the effectiveness of MARL and the global average pooling (GAP) layer.
> * Conducted a new experiment to demonstrate the effectiveness of alternating training.
> * Conducted new experiments on additional datasets to demonstrate the generalization capability of our AltO framework.
>
> If there are any further questions or clarifications needed from our side, we would be more than happy to provide them. We look forward to any feedback you might have and are eager to engage in any further discussion to improve our work.
>
> Thank you once again for your time and consideration.
>
> Best regards, Authors

---

> ### Comment · Reviewer_EsVS · 2024-08-12
>
> Thanks to the authors for the rebuttal. I have read the rebuttal and other reviewers' comments. However, I still have the following concerns:
>
> (1) In Weakness 1, my concern is mainly about why the GBT loss can enable the homography network to converge well without direct supervision of global/local motion. In my view, the GBT loss can be seen as the 'cross-modal image intensity similarity' mentioned in SCPNet, which may lead to non-convergence. Is there any theoretical basis or proof to support this?
>
> (2) Figure R2 further raises my concern about the experimental results. The visualized feature maps are not similar at all. They barely retain the structural information of the original images and contain many artifacts. I am not convinced that such feature maps can effectively supervise the registration network. What's more, in Table 2 in the paper, the author claims that simply using MSE on such feature maps (the mean of squared error of the intensity of the feature maps in Figure R2) as the Geometry loss can produce a relatively accurate result, which also reduces the credibility of the experiments.
>
> (3) Insufficient references and comparison experiments. As mentioned by reviewer sFim, this paper did not discuss many references. Moreover, comparison experiments with other methods should be conducted on all datasets to demonstrate the effectiveness of AltO, instead of only on Google Maps.
>
> (4) The homography estimation accuracy on the OS Dataset is unsatisfactory with the MACE of 14.12, which may not be regarded as a converged training. The effectiveness of the proposed method is insufficient on such cross-modal dataset.
>
> For the reasons mentioned above, I am still inclined to reject.

---

> > ### Author Response · Authors · 2024-08-13
> >
> > Thank you for your continued attention to our work. We would like to address the remaining concerns and the additional questions you have raised.
> >
> > (1) Due to the encoder’s mapping, the GBT loss is calculated under uni-modal conditions in the feature space, not cross-modal conditions in the image space. Although the feature spaces might not perfectly match, a few corresponding regions(or 4-corresponding points) are sufficient to estimate the homography. This is evident in the feature map in Figure R2, where a few corresponding regions are strongly activated. Additionally, experiments (Table R1) show that GBT alone, without MARL, fails because the encoder doesn't learn the mapping without MARL.
> >
> > (2) The purpose of the geometry loss is not to make the features identical. Even if the given pair of feature maps differ somewhat, the goal is to estimate the homography by matching the most strongly activated corresponding points. The encoder plays a crucial role in this process, ensuring that these key points are strongly activated in both input images, which makes the geometry loss convex at these points. Without the encoder, achieving convexity would be difficult when the modalities differ.
> >
> > Since the feature maps are not perfectly identical, as seen in Figure R2, the lower bound (bias) of the geometry loss may be higher. However, if at least 4-corresponding points are well-matched, the homography can still be accurately estimated. The same applies when using MSE: as long as a few strongly activated local regions are well matched, the homography can be accurately estimated even if the two feature maps are not identical. Furthermore, while MSE achieves its minimum when the two feature maps are exactly the same, the GBT loss, being based on normalized similarity, can reach its minimum as long as the trends are similar. This property makes the GBT loss more suitable for this scenario, as shown in the ablation study of our paper.
> >
> > (3) Conducting experiments for all possible cases requires significant time, so we focused on the most meaningful ones. As you mentioned, the Google Map dataset has a significant modality gap, which is why we prioritized it. However, we agree that additional experiments would strengthen our paper.
> >
> > (4) Since the OS dataset is completely new to us, the existing hyperparameters may not have been suitable. Additionally, our proposed method is a learning framework, so improvements could be made through architecture exploration of the encoder and projector, but this has not been done yet for the OS dataset. Therefore, the absolute performance may appear lower than on other datasets. Despite these challenging conditions, our method shows better performance than other baselines, further demonstrating its tendency to converge and its overall feasibility.
> >
> > We apologize for the lack of detailed explanation regarding point (2) in our paper, and we hope this clarification is helpful. If you have any further questions or concerns, please feel free to ask. We would be more than happy to provide a thorough response to any additional inquiries.

---

> > > ### Comment · Reviewer_EsVS · 2024-08-14
> > >
> > > Thank you to the authors for their response. However, it seems that some of my questions were misunderstood, and the reply does not fully address my concerns. Additionally, some statements are not well-supported and contain significant conceptual errors. Below is a point-by-point discussion:
> > >
> > > (1) The original issue (1), corresponding to Weakness 1.1 in the review and comment (1), questions how training can converge effectively under large homography deformations (e.g., [-32, 32] pixel displacement of the four corner points of a 128x128 image) when only using supervision based on content similarity, such as GBT, especially for iterative networks. Previous unsupervised works, such as Smurf [1] and MPIFlow [2], attempt to directly supervise the motion field by simulating ground truth for training iterative networks, rather than merely supervising on content similarity. Moreover, based on my understanding, the proposed GBT loss and the “cross-modal intensity-based loss” in SCPNet [3] both measure the content similarity of input images. According to experiments in SCPNet, training could not converge only under the supervision of the cross-modal intensity-based loss. Is there any theoretical basis or proof to support why the proposed GBT loss, which also measures content similarity, can enable unsupervised training to converge under such large homography deformation?
> > >
> > > Furthermore, the authors' explanation for this issue is conceptually incorrect. The authors state, "Although the feature spaces might not perfectly match, a few corresponding regions (or 4-corresponding points) are sufficient to estimate the homography". While it is possible to estimate homography with only a few matching points using manually designed algorithms like RANSAC [4], due to strong prior knowledge, the problem under discussion is the unsupervised training of homography estimation networks based on content similarity, not the design of manual homography estimation methods. The network training method proposed by the authors in the paper lacks prior knowledge like RANSAC. At the beginning of training, it can only infer the homography based on content similarity in the feature maps, making the requirement for consistency in corresponding regions significantly higher.
> > >
> > > Finally, the authors also acknowledge in Figure R2 that only "a few corresponding regions are strongly activated," and the feature map consistency shown in Figure R2 is poor. Such feature maps are unconvincing and do not guarantee the success of the proposed network training methods. Even when using manually designed homography estimation methods like RANSAC, a small number of corresponding regions could lead to poor estimation results. When corresponding regions are concentrated in a small part of the image, even a slight matching error can be magnified in the overall homography estimation. I am concerned about the authors' understanding of the fundamental concepts in this field.
> > >
> > > (2) The authors seem to misunderstand my original question. I fully acknowledge that “The purpose of the geometry loss is not to make the features identical,” and my original question did not suggest that the geometry loss needs to do so. As the authors stated in the paper (Lines 173-174), “The Modality loss aims to constrain z_A and z_B to share a similar representational space,” meaning that the consistency of the feature maps in Figure R2 is expected to be ensured by MARL.
> > >
> > > My original question was whether the Modality-Agnostic Representation Learning (MARL) process, which uses global average pooling (GAP) and thus does not supervise spatial information in the loss, might lead to poor local consistency of the constrained feature maps. In fact, Figure R2 seems to confirm this concern, as the feature map consistency is poor, with even noticeable artifacts.
> > >
> > > Additionally, I question how the proposed geometry loss can still produce good homography estimation results on such low-quality feature maps. Even more puzzling, Table 2 of the paper shows that replacing the geometry loss with MSE still achieves relatively good results. It raises the question: why wouldn’t the stripe artifacts in Figure R2 mislead homography estimation? The authors' explanation, however, does not address the core of this issue.
> > >
> > > (3) I understand that conducting more experiments may not be easy during the rebuttal period. However, since the OS dataset is more multimodal compared to the Google Map dataset, conducting additional experiments in the future is indeed necessary.
> > >
> > > (4) I agree with the authors’ explanation for the performance on the OS dataset.
> > >
> > >
> > > [1] Smurf: Self-teaching multi-frame unsupervised raft with full-image warping. In CVPR, 2021.
> > > [2] Mpi-flow: Learning realistic optical flow with multiplane images. In ICCV, 2023.
> > > [3] SCPNet: unsupervised cross-modal homography estimation via intra-modal self-supervised learning. In ECCV, 2024.
> > > [4] http://6.869.csail.mit.edu/fa12/lectures/lecture13ransac/lecture13ransac.pdf

---

> ### Author Response · Authors · 2024-08-14
>
> (1) We apologize for the misunderstanding. It seems we may have overcomplicated your question. To clarify, let’s first look at UDHN [1], the pioneering unsupervised method in a same-modality scenario, which might address most of your concerns. UDHN predicts four corresponding points, then converts them into a homography using Direct Linear Transformation (DLT). The source image is then warped using the predicted homography, and a reconstruction loss or intensity-based similarity is calculated with the target image. This entire process, including DLT and warping, is differentiable and converges very well in a same-modality scenario. A similar approach is taken by biHomE [2], and both UDHN and biHomE are incorporated into our proposed method.
>
> Our method targets multimodal datasets. Therefore, we internally convert them to a uni-modal condition to apply an approach similar to UDHN or biHomE. To facilitate this multimodal-to-uni-modal conversion, we introduced MARL and alternating training. Thus, the GBT loss is calculated under uni-modal conditions thanks to the encoder's mapping.
>
> In the aforementioned processes, although methods like RANSAC are not used, the approaches work well. It seems you might be considering feature-based frameworks like SIFT, SURF, ORB or methods like LIFT [3] and Super Point [4], which replace parts of this process with deep learning. However, our method falls within the category of end-to-end homography estimation methods, like DHN [5], UDHN, and biHomE. These methods are composed entirely of differentiable processes for end-to-end learning.
>
> Rest assured, we are fully aware of the fundamentals of feature-based frameworks and understand practical know-how, such as:
>
> * If the corresponding points are clustered in one region or many lie on a straight line, the error in homography estimation increases significantly.
> * Conversely, the homography estimation becomes more accurate when the corresponding points are widely distributed across the entire image.
> * If only a few corresponding points are known, even with methods like RANSAC, the homography estimation error can increase due to outliers.
>
> Finally, regarding Figure R2, please consider that this feature map is simply an average across 128 channels. Since our GBT is based on the Pearson correlation coefficient, the strength level (bias) difference between the two feature maps is not important; only the similarity in trends between the two feature maps reduces the loss. Additionally, both MSE and GBT, it can allow a few strongly activated regions (not just single points, but multiple points) to accurately infer the overall homography. This can be empirically observed by looking at the regression values computed by the registration network. Again, unlike explicit methods like RANSAC, the registration network implicitly calculates these values through learning.
>
> [1] Ty Nguyen et al., Unsupervised deep homography: A fast and robust homography estimation model. IEEE Robotics Autom. Lett., 2018.
>
> [2] Daniel Koguciuk et al., Perceptual loss for robust unsupervised homography estimation. In IEEE Conference on Computer Vision and Pattern Recognition Workshops, CVPR Workshops 2021.
>
> [3] YI, Kwang Moo, et al. Lift: Learned invariant feature transform. In: Computer Vision–ECCV 2016
>
> [4] DETONE, Daniel et al., Superpoint: Self-supervised interest point detection and description. In: CVPRW 2018
>
> [5] Daniel DeTone et al., Deep image homography estimation. CoRR, abs/1606.03798, 2016.
>
> (2) We now understand that your main concern is whether the encoder's output can maintain geometric consistency. We agree that the lack of an explicit mechanism could be a potential weakness, as you mentioned. However, as shown in Table R2, removing the global average pooling layer is not an option. We think that further improvements may require additional processing directly on the encoder’s output to address this issue.
>
> Additionally, regarding the discussion on MSE, please refer to the last paragraph of (1).
>
> (3) We agree that the OS dataset is more multimodal than the Google Map dataset, so further research will need to focus on this dataset.
>
> Thank you for your detailed and ongoing discussion.

---

> > ### Comment · Reviewer_EsVS · 2024-08-14
> >
> > The reasons provided by the authors are not convincing and have not addressed my concerns. Therefore, I maintain my rating.
> >
> > (1) The works cited by the authors, namely UDHN and biHomE, **neither have successful precedents of training on cross-modal data nor iterative networks like IHN, RHWF, and RAFT**, making their explanation unpersuasive.
> >
> > (2) The authors mentioned that their feature maps are averaged across 128 dimensions. First, I reviewed the original paper, and this detailed information was never mentioned. Second, to my understanding, averaging should weaken artifacts, contrary to what is shown in Figure R2, where there are many artifacts with poor consistency.
> >
> > (3) Furthermore, I have never requested a comparison with RANSAC-type algorithms. RANSAC was mentioned because the authors explained the reason for successful unsupervised network training by stating, "Although the feature spaces might not perfectly match, a few corresponding regions (or 4-corresponding points) are sufficient to estimate the homography." In my view, this explanation may have a few possibilities that make sense for RANSAC-type algorithms, **but it is highly unreasonable and unfeasible for the cross-modal unsupervised homography training adopted by the authors**, and they have not clarified this issue.

---

> ### Author Response · Authors · 2024-08-14
>
> We regret that our explanation did not fully satisfy you, but we appreciate your engagement in the discussion until the end.
>
> (1) As mentioned, UDHN and biHomE assume a uni-modal scenario. **Our main contribution lies in expanding this to a multimodal application by introducing MARL and alternating training.** Regarding the iterative framework, we retained the original structure of RAFT, IHN, and RHWF, simply replacing the ground truth with AltO. This is why we did not emphasize it in the paper, but we demonstrated that it can be applied without significant issues through the main experiments.
>
> (2) Evaluations of the visualized feature maps are subjective, so we believe further discussion on this might not be productive.
>
> (3) In our previous comment, we mentioned that "it can allow a few strongly activated regions (not just single points, but multiple points) to accurately infer the overall homography." In other words, even with only a few distinctive regions in a solid image, the registration network can capture these and perform homography estimation relatively accurately.

---

### Official Review · Reviewer_sFim · 2024-07-03

**Soundness:** 2
**Presentation:** 3
**Contribution:** 3
**Rating:** 7
**Confidence:** 4

**Summary:**

This paper proposes a new unsupervised homography estimation approach for multimodal images. This method is designed as a two-phase optimization framework named AltO. The first phase named "Geometry Learning" trains a registration network to align the input multimodal images geometrically. The second phase named "Modality-Agnostic Representation Learning" trains an encoder and a projector to extract the image-level features invariant to modality changes. Experimental results demonstrate that AltO outperforms several existing unsupervised approaches on the multimodal registration datasets.

**Strengths:**

1. The proposed framework is intuitive and interesting. This framework trains a registration network to align the input multimodal images geometrically, and trains another encoder to match the image-level features of the warped multimodal images. This framework has the potential to capture the pixel-level and image-level information in an unsupervised manner.
2. The organization and presentation of this paper are good. I think I can understand the core idea of this paper.

**Weaknesses:**

**1. Some central claims of this paper lack experimental evidence.**

1.1 The "alternating" optimization framework is a central design in this paper. However, why is "alternating" optimization necessary? Will optimizing the "geometry loss" and "modality loss" simultaneously hurt performance?

1.2 The superiority of the proposed Geometry Barlow Twins (GBT) loss was not verified. The original Barlow Twins loss can be straightforwardly applied to the proposed model by considering both the spatial axis (indexed with "h,w") and batch axis (indexed with "n") as the batch dimension. This straightforward implementation should be compared with the proposed GBT loss.

1.3 The proposed approaches should be compared with some recent unsupervised approaches. Here are some approaches with released codes.

[1] Unsupervised global and local homography estimation with motion basis learning. IEEE Transactions on Pattern Analysis and Machine Intelligence, 2022.

[2] A Multiscale Framework with Unsupervised Learning for Remote Sensing Image Registration, IEEE Transactions on Geoscience and Remote Sensing, 2022.

**2. This paper did not discuss the recent hand-crafted approaches for multimodal image registration.**

Many recent hand-crafted methods have been published in the top journals, so this kind of approach should not be ignored. The experiment should also compare the proposed approaches with the recent hand-crafted approaches. Here are some hand-crafted approaches with released code.

[3] Histogram of the orientation of the weighted phase descriptor for multi-modal remote sensing image matching. ISPRS Journal of Photogrammetry and Remote Sensing, 2023.

[4] POS-GIFT: A geometric and intensity-invariant feature transformation for multimodal images. Information Fusion, 2024.

**3. The discussion of the motivation is not sufficient.**

The Introduction section mentioned some typical unsupervised approaches designed for the images from the same modality (e.g., UDHN and biHomE). However, the unsupervised approaches [2,5] designed for multimodal image registration are not discussed. What is the motivation of the proposed method compared with this kind of approach?

[5] A Novel Coarse-to-Fine Deep Learning Registration Framework for Multi-Modal Remote Sensing Images. IEEE Transactions on Geoscience and Remote Sensing, 2023.

**4. This paper misses some references to hand-crafted and unsupervised approaches.**

I have listed some of them in the above weaknesses. The authors should further survey more papers and carefully revise the "Related Work" section.

**Questions:**

Please provide more discussions and experimental results to address the above weaknesses.

Moreover, is the 3D reconstruction task related to "Homography Estimation" (line 21)? Generally, 3D reconstruction focuses on non-planar scenes, while homography estimation is designed for the planar scenes. Is there some literature that mentions the relationship between 3D reconstruction and homography estimation?

**Limitations:**

The authors have discussed the limitations.

---

> ### Author Rebuttal · Authors · 2024-08-06
>
> Thank you for taking the time to review our work in detail. Below are our responses to your comments and concerns.
>
> **Weakness 1.**
>
> **Weakness 1.1.** *Why is alternating necessary?*
>
> To prevent unintended collaborations and a collapse into trivial solutions, we introduced an alternating training strategy. Training the encoder and registration network together end-to-end without alternating can cause the encoder to output a constant value and the registration network to always output the identity matrix as the homography. Alternating training isolates these modules. This ensures the encoder maps input images to the same feature space, and the registration network estimates the true homography between the two images.
>
> Table R1 in Auther Rebuttal (above) shows the experimental results when alternating training is not applied, demonstrating that proper training does not occur.
>
> **Weakness 1.2.** *What is the superiority of the proposed Geometry Barlow Twins (GBT)?*
>
> GBT is robust in both cases: whether the i.i.d. assumption of dataset holds or not. The method you suggested, using NHW as a batch, matches GBT's performance only when the i.i.d. assumption is satisfied. In the Google Map dataset, where the i.i.d. assumption is met, both loss functions perform similarly (Table R2). However, in the case shown in Figure R3 of the rebuttal PDF, the i.i.d. assumption is not met, and GBT performs better.
>
> The key point is whether the distribution over NHW dimensions is similar to that over HW dimensions. If the i.i.d. assumption is not met, the distribution over N distorts NHW, leading to a distribution dissimilar compared to HW dimensions.
>
> Table R2: Comparison of two types of Geo. loss on the Google Map dataset.
> | Method | NHW as batch | GBT |
> |-------:|:------------:|:---:|
> | DHN + AltO | **5.35** | 6.19 |
> | RAFT + AltO | 3.55 | **3.10** |
> | IHN + AltO | 3.16 | **3.06** |
> | RHWF + AltO | **3.37** | 3.49 |
>
> **Weakness 1.3.** & **Weakness 2.** *The proposed approaches should be compared with some recent unsupervised approaches and hand-crafted approaches.*
>
> Sorry for the insufficient number of baselines. We have added more baselines to Table R3 below. Some of the entries in Table R3 cite SCPNet [1], a very recent paper accepted at ECCV 2024. Nonetheless, **our method demonstrates even superior performance compared to SCPNet.**
>
> Additionally, the paper [2] you mentioned is designed to address the 6-DOF (degree of freedom) problem, so it cannot be applied to our dataset, which has 8-DOF. Furthermore, the code for that paper is not fully available. Moreover, we also attempted to run the code for paper [3], but encountered errors with the sample images, so we could not include it in our results.
>
> Table R3: MACE evaluation results on the Google Map dataset. * indicates values reported from SCPNet [1].
> | Type | Method | MACE |
> |:----:|:-------|:----:|
> | Hand-crafted | SIFT | 24.53* |
> | | ORB | 24.52* |
> | | DASC | 21.76* |
> | | RIFT | 16.55* |
> | | POS-GIFT | 20.90 |
> | Unsupervised | UDHN | 28.58 |
> | | CA-UDHN | 24.00 |
> | | biHomE | 24.08 |
> | | BasesHomo | 24.49* |
> | | UMF-CMGR | 24.60* |
> | | SCPNet | 4.35* |
> | | IHN + AltO (Ours) | **3.06** |
>
> [1] Runmin Zhang et al., SCPNet: Unsupervised Cross-modal Homography Estimation via Intra-modal Self-supervised Learning, (ECCV 2024 Accepted.)
>
> [2] Yuanxin Ye et al., A Multiscale Framework with Unsupervised Learning for Remote Sensing Image Registration, IEEE Transactions on Geoscience and Remote Sensing, 2022.
>
> [3] ZHANG et al., Histogram of the orientation of the weighted phase descriptor for multi-modal remote sensing image matching. ISPRS Journal of Photogrammetry and Remote Sensing, 2023.
>
> **Weakness 3.** *The discussion of the motivation is not sufficient.*
>
> The problem we aim to solve is the unsupervised learning of 8-DOF homography estimation for multimodal image pairs. These conditions are very common in many fields, such as industry. For example, matching a real photo to a CAD image. At the time of our research, we found very few papers that directly addressed this specific problem. This scarcity of relevant work highlighted the difficulty and value of addressing this challenge, which motivated us to pursue it.
>
> The papers you suggested, [2] and [4], do not directly apply: [2] focuses on a 6-DOF problem, and [4] employs a supervised learning approach.
>
> [4] A Novel Coarse-to-Fine Deep Learning Registration Framework for Multi-Modal Remote Sensing Images. IEEE Transactions on Geoscience and Remote Sensing, 2023.
>
> **Weakness 4.** *This paper misses some references to hand-crafted and unsupervised approaches.*
>
> We apologize for missing them. We will include and refer them during the revision.
>
> **Question 1.** *Is there some literature that mentions the relationship between 3D reconstruction and homography estimation?*
>
> Below are several documents regarding your question. We will revise the citation section related to 3D reconstruction in our paper.
>
> * Zhang, Zhongfei, and Allen R. Hanson. "3D reconstruction based on homography mapping." Proc. ARPA96 (1996): 1007-1012.
> * Mei, Christopher, et al. "Efficient homography-based tracking and 3-D reconstruction for single-viewpoint sensors." IEEE Transactions on Robotics 24.6 (2008): 1352-1364.
> * https://github.com/ziliHarvey/Homographies-for-Plane-Detection-and-3D-Reconstruction
> * Yong-In, Yoon, and Ohk Hyung-Soo. "3D reconstruction using the planar homograpy." The Journal of Korean Institute of Communications and Information Sciences 31.4C (2006): 381-390.
> * Zhang, Beiwei, and Y. F. Li. "An efficient method for dynamic calibration and 3D reconstruction using homographic transformation." Sensors and Actuators A: Physical 119.2 (2005): 349-357.
> * Dubrofsky, Elan. "Homography estimation." Diplomová práce. Vancouver: Univerzita Britské Kolumbie 5 (2009).

---

> > ### Comment · Reviewer_sFim · 2024-08-12
> >
> > Thank the authors for the responses. The additional experimental results and discussions address my main concerns. In my opinion, the ablation study about the alternating optimization and the comparison with the recent approaches make the proposed approach's superiorities more convincing. Therefore, I’d like to raise my rating from "Weak Accept" to "Accept".
> >
> > It would be better to discuss the following two problems further.
> > 1. In the response to W1.1, the authors pointed out that "Training the encoder and registration network together end-to-end without alternating can cause the encoder to output a constant value and the registration network to always output the identity matrix as the homography." It would be better to provide some intuitive explanations. Why does the end-to-end training make the encoder/registration network tend to output a constant value/identity matrix?
> > 2. In the response to W3, the authors claimed that this paper’s motivation is different from the literature [2] because the proposed approach considers the 8-DOF homography estimation while the method [2] focuses on a 6-DOF problem. However, such a difference seems to be minor because both the two methods utilize learnable regression models. It is straightforward to extend the method [2] to handle 8-DOF homography estimation. More intrinsic differences should be discussed to highlight the motivation of this paper.

---

> ### Author Response · Authors · 2024-08-12
> **Looking forward to your post-rebuttal comment!**
>
> Dear Reviewer sFim
>
> Thank you once again for participating in the review process and for providing such thoughtful feedback. We wanted to kindly follow up regarding the rebuttal we submitted. We understand that this is a busy time, and we greatly appreciate your efforts. To summarize our rebuttal to your review:
>
> * Conducted a new experiment to demonstrate the effectiveness of alternating training and MARL.
> * Conducted a new experiment and presented a case to compare GBT with the new method you suggested.
> * Added more baselines for performance evaluation by conducting a new experiment and incorporating reported values.
> * Provided detailed answers to all the questions you raised.
>
> If there are any further questions or clarifications needed from our side, we would be more than happy to provide them. We look forward to any feedback you might have and are eager to engage in any further discussion to improve our work.
>
> Thank you once again for your time and consideration.
>
> Best regards, Authors

---

> ### Author Response · Authors · 2024-08-12
> **Thank you for increasing your score!**
>
> Thank you very much for raising the rating of our paper. We appreciate your recognition of the additional experiments and discussions we provided, and we are glad that they addressed your main concerns.
>
> We would like to provide further clarification:
>
> 1. Intuitive Explanation for End-to-End Training without Alternating:
>
> If the encoder and registration network collapse as we mentioned, then all reconstruction-based losses, including L1-loss or our GBT loss, would become zero, the minimum possible value for such losses. This occurs because the encoder's outputs are constant, resulting in the difference or similarity always converging to a trivial value, regardless of the homography. Consequently, the registration network would not make any effort to find the true homography, outputs only identity matrix.
>
> 2. Difference Between Our Approach and Literature [2]:
>
> As you mentioned, extending from 6-DOF to 8-DOF is indeed straightforward, but it is a more challenging task, and some performance degradation is to be expected. The CFOG, used in [2], seems to be a key element. The multi-scale framework is aimed at more precise estimation, but it does not address different modalities. Additionally, since the code is not fully available at the moment, quickly reproducing and modifying it is difficult. However, we fully agree that the extended version of [2] would certainly be a valuable baseline to consider.
>
> We hope these clarifications address your questions, and we are more than happy to discuss further if needed. Thank you once again for your valuable insights and support.

---

### Official Review · Reviewer_PrYW · 2024-07-12

**Soundness:** 3
**Presentation:** 3
**Contribution:** 3
**Rating:** 6
**Confidence:** 5

**Summary:**

The paper addresses unsupervised homography estimation from multi-modal image pairs. The authors propose to cope with the issue of 1) modality, 2) registration in two distinct networks that are trained in an interleaved fashion. The networks architecture derives from the Barlow Twins framework, with changes in the loss function. Results are illustrated on several public benchmark of small images (128^2) and compares favorably wrt to related unsupervised approach.

**Strengths:**

1- I enjoy reading the paper. I walked through the paper, first with curiosity and skepticism, then with strong interest. The approach is intuitive (adjust the two representations then compute the transformation) and compelling. I am somehow surprised that it works :) The constrastive-like loss used in  Barlow Twins  is contributing much for the network to learn the correct solution.

2- Overall, the authors are tackling an important problem (unsupervised learning) for which an original solution is proposed --while based on previous recent work. The methodology is clearly presented. Results are convincing (thought only on small images 128x128) and illustrated on various modality pairs. Quantitative results show improvement wrt related unsupervised work

**Weaknesses:**

1- Not a weakness, but a points which could have been discussed: why not simply transforming the inputs into edge maps before learning a matching/homography function (and putting aside the modality discrepancy). It would not be a very fancy approach, but I believe it could be a baseline for comparison.

2- The approach would be more convincing if each of the two modules (GL and MARL) had demonstrated their effectiveness also individually (ie same image pair modality using only GL).

**Questions:**

-  What is the size of the embedding? What is the training time? Are the Barlow Twins trained from scratch?

- Illustration seems to show  strong geometric features (ie lines) in the input images. Is it a strong limitation of the approach?

**Limitations:**

From a practical point of view, the size of the image and the strong overlap between the pairs show that the work need to be further developped for applications at full scale.
From a methodology point of view, the authors have discussed the limitation of having two networks trained in an interleaved way, with potential collision or collapse.

---

> ### Author Rebuttal · Authors · 2024-08-06
>
> Thank you for your detailed review and for taking the time to provide your feedback. Below is our rebuttal.
>
> **Weakness 1.** *Why not use an edge-based approach as a baseline?*
>
> Edge-based approaches have limitations when used as baselines with different modalities. When dealing with two images of different modalities, one image often has rich edges while the other does not. Even within the closed modality, such as day and night photos, edge differences can arise due to shadows. Thus, converting both images to edge maps and comparing them might not achieve high performance. Although paper [1] utilizes edges in its learning process, it is limited to using image pairs of exactly the same modality.
>
> Below Table R2 shows the results of applying UDHN [2], a simple unsupervised method that assumes the same modality, to edge maps converted from the Google Map dataset. The results indicate poor performance.
>
> Table R2: Evaluation result of applying edge maps to UDHN
> | Method | MACE (Google Map) |
> |--------|:----:|
> | IHN + AltO | **3.06** |
> | UDHN | 28.58 |
> | UDHN + Edge Map + L1 loss | 24.15 |
> | UDHN + Edge Map + cos loss | 24.00 |
>
> [1] Feng, Xiaomei & Jia et al., Edge-Aware Correlation Learning for Unsupervised Progressive Homography Estimation, in IEEE Transactions on Circuits and Systems for Video Technology, 2023.
>
> [2] Ty Nguyen et al., Unsupervised deep homography: A fast and robust homography estimation model. IEEE Robotics Autom. Lett., 2018.
>
> **Weakness 2.** *Demonstrate the effectiveness of each modules, GL and MARL.*
>
> The GL module enhances geometry alignment by increasing the similarity between local correspondences. It is inspired by biHomE [3], with the only differences being that biHomE uses an ImageNet-pretrained encoder which is frozen, and applies triplet loss as the loss term. The effectiveness of this approach has already been demonstrated in the paper [3] using the S-COCO dataset, which has the same modality.
>
> The role of the MARL module is to train the encoder to map input images to the same feature space. This properly trained encoder allows our geometry loss to function correctly regardless of modality differences. Table R1 in the Author Rebuttal (above) shows the results of training on the Google Map dataset without the MARL module or with the MARL module but no alternating. The result indicates that proper learning did not occur.
>
> In summary, for successful training, the GL module, MARL module, and alternating training are all essential components.
>
> [3] Daniel Koguciuk et al., Perceptual loss for robust unsupervised homography estimation. In IEEE Conference on Computer Vision and Pattern Recognition Workshops, CVPR Workshops 2021.
>
> **Question 1.**
>
> **Question 1.1.** *What is the size of the embedding?*
>
> AltO serves as a learning framework that does not impose restrictions on the internal structure of each component. However, the encoder and projector used as examples in our paper are based on ResNet-34. The encoder uses the first two stages of ResNet-34, resulting in an embedding size of 128. Nevertheless, these can be flexibly replaced if needed.
>
> **Question 1.2.** *What is the training time?*
>
> Table R3 below shows the training time measurements on Google Map dataset. Although the training time increases, the inference time at run-time is the same as that of supervised learning since the AltO module is not needed.
>
> Table R3: Training times for each registration network. (Nvidia RTX 8000)
> |     | Supervised. | Unsupervised (with AltO) |
> |:-----:|:-----------:|:------------------------:|
> |DHN  | 1h 26m      | 4h 31m                   |
> |IHN  | 3h 27m      | 9h 57m                   |
>
> **Question 1.3.** *Are the Barlow Twins trained from scratch?*
>
> There are no pretrained components. All layers are trained from scratch.
>
> **Question 2.** *Are strong geometric features (i.e., lines) necessary for the proposed method?*
>
> It is evident that lines, such as edges, are helpful. However, as visualized in Figure R2 of the rebuttal PDF, the output of the encoder shows strong features in the form of blobs rather than edges. This indicates that not only lines but also blobs play a significant role in the proposed method.
>
> **Limitation.** *Small image size and strong overlapped pairs*
>
> The dataset settings we used (such as size and overlap) are based on the pioneering paper DHN [4] on end-to-end homography estimation. These settings have become the standard and are used by many studies, including [1-7]. For fair comparison, we followed these settings. However, AltO, as a learning framework, has no size or overlap restrictions. Constraints, if any, come from the registration network (e.g., IHN), encoder, or projector.
>
> Table R4 shows experiments where the Google Map dataset size and displacement were doubled, called 'Google Map x2.' We used DHN as the registration network, which is flexible with size constraints. The results show that DHN with AltO can reduce MACE even without any hyperparameter tuning.
>
> Table R4: MACE evaluation results on Google Map x2.
> | Method | MACE (Google Map x2) |
> |--------|:----------------------:|
> | (No warping) | 49.33 |
> | DHN + AltO | **34.24** |
>
> [4] Daniel DeTone et al., Deep image homography estimation. CoRR, abs/1606.03798, 2016.
>
> [5] Yiming Zhao et al., Deep lucas-kanade homography for multimodal image alignment. In IEEE Conference on Computer Vision and Pattern Recognition, CVPR 2021.
>
> [6] Si-Yuan Cao et al., Iterative deep homography estimation. In IEEE/CVF Conference on Computer Vision and Pattern Recognition, CVPR 2022.
>
> [7] Si-Yuan Cao et al., Recurrent homography estimation using homography-guided image warping and focus transformer. In IEEE/CVF Conference on Computer Vision and Pattern Recognition, CVPR 2023.

---

> ### Author Response · Authors · 2024-08-12
> **Looking forward to your post-rebuttal comment!**
>
> Dear Reviewer PrYW
>
> Thank you once again for participating in the review process and for providing such thoughtful feedback. We wanted to kindly follow up regarding the rebuttal we submitted. We understand that this is a busy time, and we greatly appreciate your efforts. To summarize our rebuttal to your review:
>
> * Conducted a new experiment to demonstrate the performance of the edge-based method in multimodal case.
> * Conducted a new experiment to show the effectiveness of alternating training and MARL.
> * Provided detailed answers to all the questions you raised.
> * Conducted a new experiment to demonstrate that our method, AltO, is also effective for larger image set.
>
> If there are any further questions or clarifications needed from our side, we would be more than happy to provide them. We look forward to any feedback you might have and are eager to engage in any further discussion to improve our work.
>
> Thank you once again for your time and consideration.
>
> Best regards, Authors

---

> ### Author Response · Authors · 2024-08-13
> **Thanks to Reviewer PrYW**
>
> We would like to thank you again for highly appreciating the strengths of our work.
>
> If you have any further questions or concerns, please feel free to ask. We would be more than happy to provide a thorough response to any additional inquiries.
>
> Best regards, Authors

---

> > ### Comment · Reviewer_PrYW · 2024-08-13
> >
> > Thank you to the authors for the responses and clarifications, and overall rebuttal.
> > - Comments regarding the use of edge maps make sense, I appreciate the additional experimental results which support the observations of the authors.
> > - My question was about confirming the viability of each of the two networks (MARL and GL) by testing then independently:  experiments that would be using same modality inputs (with some deformation) to evaluate GL independently of MARL. (I do not see how MARL could be tested independently of GL);
> >
> > Overall, the authors mainly addressed my concerned. Additional experiments have consolidated the work. I keep my score as accept.

---

> > > ### Author Response · Authors · 2024-08-14
> > >
> > > Thank you for your continued feedback. We noticed that there may have been a misunderstanding regarding one of the points you raised. We would like to clarify this and provide a more accurate response.
> > >
> > > We have conducted the following additional experiments using the MS-COCO dataset, which has the exact same modality. Due to the limited time, we used the simple DHN model as the registration network. In the case where only GL is used, the encoder was frozen from scratch.
> > >
> > > | Method | MACE |
> > > |------------|:--------:|
> > > | (No warping ) | 24.89 |
> > > | DHN + GL | 5.75 |
> > > | DHN + MARL | 25.00 |
> > >
> > > In the case where only MARL is used, the spatial information is lost due to the global average pooling layer, leading to ineffective learning.
> > >
> > > We hope this response addresses your concerns. If you have any further questions, please feel free to ask. We will do our best to respond promptly.
> > >
> > > Thank you.

---

### Author Rebuttal · Authors · 2024-08-06

We sincerely thank the reviewers for taking the time to thoroughly review our paper.

Reviewers highlighted the strengths of our paper.

+ Proposed method is an interesting, intuitive, and fresh approach. (PrYW, sFim, EsVS)
+ The paper tackles an important problem and proposes an original solution. (PrYW)
+ Quantitative results show improvement. (PrYW)
+ The organization and presentation of this paper are clear and good. (PrYW, sFim)

We will incorporate the reviewers' advice to further refine our research and improve the paper.

Additionally, through the review process, we found that our method outperforms the very recent paper, *SCPNet: Unsupervised Cross-modal Homography Estimation via Intra-modal Self-supervised Learning* (ECCV 2024 accepted).

The following experimental result was obtained using the Google Map dataset. * indicates values reported from SCPNet. (Lower is better.)
| Type | Method | MACE |
|:----:|:-------|:----:|
| Hand-crafted | SIFT | 24.53* |
| | ORB | 24.52* |
| | DASC | 21.76* |
| | RIFT | 16.55* |
| | POS-GIFT | 20.90 |
| Unsupervised | UDHN | 28.58 |
| | CA-UDHN | 24.00 |
| | biHomE | 24.08 |
| | BasesHomo | 24.49* |
| | UMF-CMGR | 24.60* |
| | SCPNet | 4.35* |
| | IHN + AltO (Ours) | **3.06** |

--------------------------------------------------------------------------------------------------------------------------------

Note :
* The attached PDF includes figures addressing each reviewer's comments.

* Table R1 below shows the result of the experiment addressing the common question, 'Why is alternating necessary?'

Table R1: Ablation result on the Google Map dataset for applying the alternating and MARL module. (Alt. is Alternating)
|  **Method** | **No Alt.(with MARL)** | **No Alt.(without MARL)** | **Alt.(with MARL)** |
|:-----------:|:----------------------:|:-------------------------:|:-------------------:|
|  DHN + AltO |          24.09         |           24.27           |       **6.19**      |
| RAFT + AltO |          26.21         |           25.91           |       **3.10**      |
|  IHN + AltO |          24.37         |           23.14           |       **3.06**      |
| RHWF + AltO |          18.88         |           24.07           |       **3.49**      |

---

### Decision · Program_Chairs · 2024-09-25

**Decision:**

Accept (poster)

**Comment:**

The reviewer opinion is split, even after seeing each others reviews and after an extensive post-rebuttal discussion. The main strength of the paper is the problem setting:  the handling of unsupervised training for a geometric problem with a significant domain gap, addressed by a novel loss function  The critical reviewer lists problems that are mainly technical in nature, but none of them is "rejection-implying". The rebuttal covered some of the issues (lack of references, same experiments that provide clarification).

The paper is close to the accept - reject boundary. Overall, we see a benefit  in presenting the method at the conference, despite some remaining technical issues.